

# Multiple comparisons of precipitation variations in different areas using simultaneous confidence intervals for all possible ratios of variances of several zero-inflated lognormal models

Patcharee Maneerat[1] and Sa-Aat Niwitpong[2]

[1] Department of Applied Mathematics, Rajabhat Uttaradit University, Uttaradit, Thailand
[2] Department of Applied Statistics, King Mongkut's University of Technology North Bangk, Bangkok, Thailand

## ABSTRACT

Flash flooding and landslides regularly cause injury, death, and homelessness in Thailand. An advancedwarning system is necessary for predicting natural disasters, and analyzing the variability of daily precipitation might be usable in this regard. Moreover, analyzing the differences in precipitation data among multiple weather stations could be used to predict variations in meteorological conditions throughout the country. Since precipitation data in Thailand follow a zero-inflated lognormal (ZILN) distribution, multiple comparisons of precipitation variation in different areas can be addressed by using simultaneous confidence intervals (SCIs) for all possible pairwise ratios of variances of several ZILN models. Herein, we formulate SCIs using Bayesian, generalized pivotal quantity (GPQ), and parametric bootstrap (PB) approaches. The results of a simulation study provide insight into the performances of the SCIs. Those based on PB and the Bayesian approach via probability matching with the beta prior performed well in situations with a large amount of zero-inflated data with a large variance. Besides, the Bayesian based on the reference-beta prior and GPQ SCIs can be considered as alternative approaches for small-to-large and medium-to-large sample sizes from large population, respectively. These approaches were applied to estimate the precipitation variability among weather stations in lower southern Thailand to illustrate their efficacies.

## INTRODUCTION AND MOTIVATION

In early 2021, approximately 186,300 people in lower southern Thailand were affected by heavy rainfall resulting in flash flooding, landslides, and windstorms, as reported by Thailand's Department of Disaster Prevention and Migration (DDPM) (*Thailand, 2021*). Four provinces in the lower southern region of Thailand were affected by flooding: Songkhla (60 households), Pattani (2,810 households), Yala (12,082 households), and

Corresponding author
Sa-Aat Niwitpong,
sa-aat.n@sci.kmutnb.ac.th

Narathiwat (22,308 households). Meanwhile, landslides occurred in Yala and Narathiwat that affected approximately 57 households (*Thailand, 2021*). Unfortunately, these natural disasters resulted in deaths and injuries (*David, 2021*).

It would be possible to reduce the impact of natural disasters if governmental organizations had an early warning system that could be triggered to warn people in high-risk areas in advance of impending catastrophes. Rainfall dispersion data can provide essential information indicating imminent flooding when variation is high by analyzing historical precipitation data. Importantly, it could also be used to predict precipitation variation in each area. From the historical evidence of flooding in lower southern Thailand, the precipitation data in four areas are inflated with zero observations, while the non-zero precipitation records are log-normally distributed, as can be seen in An Empirical Application Section. These properties indicate that precipitation data obey the assumptions for a zero-inflated lognormal (ZILN) distribution and can be modeled accordingly.

The ZILN model, also referred to as the delta-lognormal model, is appropriate for modeling right-skewed data with a proportion of zero (*Aitchison & Brown, 1963*; *Fletcher, 2008*; *Wu & Hsieh, 2014*; *Hasan & Krishnamoorthy, 2018*; *Maneerat, Niwitpong & Niwitpong, 2019*). Variance is a dispersion measure of probability used in statistical inference for both point and interval (*e.g.*, confidence interval: CI) estimation. Several researchers have formulated point and interval estimates *via* various approaches. For example, *Burdick & Graybill (1984)* established CIs for linear combinations of the variance components using the unbalanced one-way classification model and the Graybill-Wang procedure by considering the inequality of the design (*Graybill & Wang, 1980*). *Ciach & Krajewski (1999)* estimated the radar-raingauge difference variances which can be separated into the area-point ground raingauge originating from resolution difference between them, and the error of the radar area-average rainfall estimate. Another important approach for variance estimation is bootstrapping based on t-statistics to formulate nonparametric CIs for a single variance and the difference between variances, which was used to estimate the variance in insurance data for properties (*Cojbasic & Tomovic, 2007*). *Bebu & Mathew (2008)* used a modified single log-likelihood ratio procedure to construct CIs for the ratio of bivariate lognormal variances and applied it to compare variation in health care costs. *Cojbasic & Loncar (2011)* suggested Hall's bootstrapped-t method for constructing one-sided CIs (lower and upper endpoint CIs) for the variances of skewed distributions and illustrated the efficacy of their method by analyzing revenue variability within the food retail industry.

Later, *Herbert et al. (2011)* suggested an analytical method for the difference between two independent variances that performed well even with small unequal sample sizes and highly skewed leptokurtic data; they used data from a randomized trial for a cholesterol-lowering drug to portray the efficacies of their proposed methods. *Harvey & Merwe (2012)* revealed that a Bayesian CI based on the highest posterior density outperformed one based on the equal-tailed interval for the variance of lognormal distribution with zero observations. *Maneerat, Niwitpong & Niwitpong (2020)* showed that the highest posterior density interval based on a probability matching prior produced the narrowest interval with correct coverage for comparing delta-lognormal variances; they applied it to estimate the

difference between rainfall variability in the lower and upper northern regions of Thailand. Recently, Bayesian credible intervals based on a non-informative prior were presented by *Maneerat, Niwitpong & Niwitpong (2021a)* for the single variance of a delta-lognormal model that was used on daily rainfall records.

Nevertheless, no studies have yet been conducted on simultaneous CIs (SCIs) for pairwise comparisons of the variances of several ZILN models, and so we addressed our research toward filling this gap. Hence, we estimated all possible ratios of variances of several ZILN models by using SCIs based on Bayesian, parametric bootstrap (PB), and generalized pivotal quantity (GPQ) approaches. The reasons for choosing them are that the Bayesian and PB approaches can be used to construct CIs capable of handling situations with large differences in the variances and high proportion of zero values of delta-lognormal models, respectively (*Maneerat, Niwitpong & Niwitpong, 2020*), while CI based on the GPQ approach perform quite well when the variance was large maneeratEstimatingFishDispersal2020. Their efficacies were determined *via* simulation studies and precipitation data from four areas of the lower southern region of Thailand in terms of the coverage rate (CR), the lower error rate (LER), the upper error rate (UER), and the average width (AW).

## MODEL AND METHODS

### Model

For $h$ groups, $d_i; i = 1, 2, \ldots, h$, denotes the probability of having zero observations while the remaining probability for non-zero observations, $d_i' = 1 - d_i$, follows a lognormal distribution denoted as $\mathrm{LN}(\mu_i, \sigma_i^2)$ with mean $\mu_i$ and variance $\sigma_i^2$. For random samples from the groups, let $Y_i = (Y_{i1}, Y_{i2}, \ldots, Y_{in_i})$ denote a ZILN variate based on $n_i$ observations from group $i$ with the probability density function given by

$$g(y_i; d', \mu_i, \sigma_i^2) = d_i + d_i' y^{-1} (2\pi\sigma_i^2)^{-1/2} \exp\left\{-\frac{(\ln y_i - \mu_i)^2}{2\sigma_i^2}\right\}. \tag{1}$$

For $Y_i = 0$, the number of zero observations $n_{i0}$ follows a binomial distribution with sample size $n_i$ and the probability of having zero observations $d_i$, where $n_i = n_{i0} + n_{i1}$, $n_{i0} = \#\{j : Y_{ij} = 0\}$ and $n_{i1} = \#\{j : Y_{ij} > 0\}$; $j = 1, 2, \ldots, n_i$. For $Y_i > 0$, $W_i = \ln Y_i$ are normally distributed with mean $\mu_i$ and variance $\sigma_i^2$. For a ZILN model, the maximum likelihood estimates of $d_i$, $\mu_i$ and $\sigma_i^2$ are $\hat{d}_i = n_{i0}/n_i$, $\hat{\mu}_i = \sum_{j:Y_{ij}>0} \ln Y_{ij}/n_{i1}$ and $\hat{\sigma}_{i,mle}^2 = \sum_{j:Y_{ij}>0} [\ln Y_{ij} - \hat{\mu}_i]^2/n_{i1}$, respectively. For the $i$th group, the population variance of $Y_i$ is given by

$$V_i = d_i' \exp(2\mu_i + \sigma_i^2)[\exp(\sigma_i^2) - d_i'] \tag{2}$$

which can be log-transformed as $T_i = \ln V_i = \ln d_i' + 2(\mu_i + \sigma_i^2) + \ln[1 - \frac{d_i'}{\exp(\sigma_i^2)}]$. Considering the third term of $T_i$ leads to obtaining $\lim_{\sigma_i^2 \to \infty} \ln[1 - \frac{d_i'}{\exp(\sigma_i^2)}] = 0$ when $\sigma_i^2$ is large. Thus, the log-transformed variance of $V_i$ can be approximated as

$$T_i \approx \ln d_i' + 2(\mu_i + \sigma_i^2). \tag{3}$$

Given $\hat{d}_i$, $\hat{\mu}_i$ and $\hat{\sigma}_i^2$ from the observations, the estimates of $T_i$ can be written as $\hat{T}_i \approx \ln \hat{d}_i' + 2(\hat{\mu}_i + \hat{\sigma}_i^2)$; $\hat{\sigma}_i^2 = \sum_{j:y_{ij}>0}[\ln Y_{ij} - \hat{\mu}_i]^2/(n_{i1} - 1)$. Using the delta theorem, the variance of $\hat{T}_i$ becomes

$$\text{Var}(\hat{T}_i) = \frac{1 - d_i'}{n_i d_i'} + 4\left(\frac{\sigma_i^2}{n_{i1}} + \frac{2\sigma_i^2}{n_{i1} - 1}\right). \tag{4}$$

In the present study, the parameter of interest is all pairwise ratios among the log-transformed variances of several ZILN models, which is defined as

$$\lambda_{ik} = \ln\left(\frac{V_i}{V_k}\right) = T_i - T_k. \tag{5}$$

Its estimates can be obtained as $\hat{\lambda}_{ik} = \hat{T}_i - \hat{T}_k$; $\forall i \neq k$ and $i, k = 1, 2, \ldots, h$. From Eq. (4), the variance of $\hat{\lambda}_{ik}$ can be expressed as

$$\text{Var}(\hat{\lambda}_{ik}) = \text{Var}(\hat{T}_i) + \text{Var}(\hat{T}_k), \tag{6}$$

where the covariance between $\hat{T}_i$ and $\hat{T}_k$ is $\text{COV}(\hat{T}_i, \hat{T}_k) = 0$ because $Y_i = (Y_{i1}, Y_{i2}, \ldots, Y_{in_i})$ comprise independent and identically distributed (iid) random vector from a ZILN model. Thus, we can obtain estimates of $\hat{T}_i$ that are independent random variables. Using estimates $(\hat{d}_i', \hat{\mu}_i, \hat{\sigma}_i^2)$ and $\hat{d}_k', \hat{\mu}_k, \hat{\sigma}_k^2$ from the samples enables the estimated variance of $\hat{\lambda}_{ik}$ to become

$$\widehat{Var}(\hat{\lambda}_{ik}) = \frac{1 - \hat{d}_i'}{n_i \hat{d}_i'} + \frac{1 - \hat{d}_i'}{n_k \hat{d}_k'} + 4\left(\frac{\hat{\sigma}_i^2}{n_{i1}} + \frac{2\hat{\sigma}_i^2}{n_{i1} - 1} + \frac{\hat{\sigma}_k^2}{n_{k1}} + \frac{2\hat{\sigma}_k^2}{n_{k1} - 1}\right), \tag{7}$$

where $(\hat{d}_i', \hat{\mu}_i, \hat{\sigma}_i^2)$ and $(\hat{d}_k', \hat{\mu}_k, \hat{\sigma}_k^2)$ denote the estimated parameters of $(d_i', \mu_i, \sigma_i^2)$ and $(d_k', \mu_k, \sigma_k^2)$, respectively.

## Methods

To estimate $\lambda_{ik}$, the SCIs are formulated based on Bayesian, GPQ and PB approaches.

### The Bayesian approach

The essential feature of Bayesian approach is to use the situation-specific prior distribution that reflects knowledge or subjective belief about the parameter of interest; this is modified in accordance with Baye's Theorem to yield the posterior distribution. Thus, CIs based on the Bayesian approach are derived by using the posterior distribution. In Bayesian theory, the CI is referred to as the credible interval because it is not unique on the posterior distribution. The following methods are used to define suitable credible intervals: the narrowest interval for a univariate distribution (the highest posterior density interval) (*Box & Tiao, 1973*); the interval when the probability of being below is the same as being above, which is sometimes referred to as the equal-tailed interval (*Gelman et al., 2014*); or the interval with the mean as the central point (assuming that it exists). In the present study, the SCIs based on the Bayesian approach were constructed based on the equal-tailed interval. Motivated by *Maneerat, Niwitpong & Niwitpong (2020)*, the probability-matching-beta (PMB) and reference-beta (RB) priors were our choice for parameter $(d_i', \mu_i, \sigma_i^2)$ in this study. Thus, Bayesian SCIs for $\lambda_{ik}$ were established as follows:

*The PMB prior:*

The probability-matching prior for $(\mu_i, \sigma_i^2)$ is $P(\mu_i, \sigma_i^2)_{pm} \propto \sigma_i^{-2}\sqrt{2 + \sigma_i^{-2}}$ combined with the prior of $d_i'$ as a beta distribution with $a_i = b_i = 1/2$. Thus, the PMB prior for $(d', \mu_i, \sigma_i^2)$ can be defined as

$$P(d', \mu_i, \sigma_i^2)_{\text{pmb}} \propto \prod_{i=1}^{h} \sigma_i^{-2} \sqrt{\frac{2 + \sigma_i^{-2}}{(1 - d_i')d_i'}}. \tag{8}$$

When updated with its likelihood, we obtain

$$P(y|\lambda) \propto \prod_{i=1}^{h} (1 - d_i')^{n_{i0}} d_i'^{n_{i1}} (\sigma_i^2)^{-n_{i1}/2} \exp\left\{ -\frac{1}{2\sigma_i^2} \sum_{j:x_{ij}>0} (\ln y_{ij} - \mu_i)^2 \right\}. \tag{9}$$

The respective marginal posterior distributions of $(d_i', \mu_i, \sigma_i^2)$ are

$$P(d_i'|y_i)_{\text{pmb}} \quad \propto (1 - d_i')^{n_{i0}+1/2} d_i'^{n_{i1}+1/2} \tag{10}$$

$$P(\mu_i|y_i, \sigma_i^2)_{\text{pmb}} \propto \exp\left\{ -\frac{1}{2\sigma_{i,\text{pmb}}^2} \sum_{j:x_{ij}>0} (\ln y_{ij} - \mu_i)^2 \right\} \tag{11}$$

$$P(\sigma_i^2|y_i)_{\text{pmb}} \quad \propto (\sigma_i^2)^{-\frac{n_{i1}+1}{2}} \sqrt{2 + \sigma_i^{-2}} \exp\left\{ -\frac{(n_{i1}-1)\hat{\sigma}_i^2}{2\sigma_i^2} \right\} \tag{12}$$

which are denoted as $d_{i,\text{pmb}}^{(post)}|y_i \sim \text{beta}(n_{i0} + 1/2, n_{i1} + 1/2)$, $\mu_{i,\text{pmb}}^{(post)}|y_i \sim \text{N}(\hat{\mu}_{i,\text{pmb}}, \sigma_{i,\text{pmb}}^{2(post)})$, and $\sigma_{i,\text{pmb}}^{2(post)} \propto (\sigma_i^2)^{-\frac{n_{i1}+1}{2}} \sqrt{2 + \sigma_i^{-2}} \exp\left\{ -\frac{(n_{i1}-1)\hat{\sigma}_i^2}{2\sigma_i^2} \right\}$, respectively. Thus, the posterior of $\lambda$ becomes

$$\lambda_{ik,\text{pmb}}^{(post)} = T_{i,\text{pmb}}^{(post)} - T_{k,\text{pmb}}^{(post)}, \tag{13}$$

where $T_{i,\text{pmb}}^{(post)} \approx \ln d_{i,\text{pmb}}^{(post)} + 2(\mu_i^{(post)} + \sigma_{i,\text{pmb}}^{2(post)})$ and $T_{k,\text{pmb}}^{(post)} \approx \ln d_{k,\text{pmb}}^{(post)} + 2(\mu_k^{(post)} + \sigma_{k,\text{pmb}}^{2(post)})$. In agreement with *Ganesh (2009)*, the $100(1 - \alpha)\%$ Bayesian-based SCI with PMB prior for $\lambda_{ik}$ is

$$[L_{\lambda_{ik}}, U_{\lambda_{ik}}]_{\text{pmb}} = \left[ \lambda_{ik,\text{pmb}}^{(post)} \mp v_\alpha^{\text{pmb}} \right], \tag{14}$$

where $v_\alpha^{\text{pmb}}$ stands for the $(1 - \alpha)^{th}$ percentile of the distribution of $V^{\text{pmb}} = \max_h \left\{ \lambda_{ik,\text{pmb}}^{(post)} \right\} - \min_h \left\{ \lambda_{ik,\text{pmb}}^{(post)} \right\}$.

*The RB prior:*

This is a non-informative prior derived from the Fisher information matrix (*Maneerat, Niwitpong & Niwitpong, 2020*). The RB prior of $(d', \mu_i, \sigma_i^2)$ is defined as

$$P(d', \mu_i, \sigma_i^2)_{\text{rfb}} \propto \prod_{i=1}^{h} \sigma_i^{-1} \sqrt{\frac{1 + (2\sigma_i^2)^{-1}}{(1 - d_i')d_i'}} \tag{15}$$

in which the prior of $d'$ is a beta distribution. When combined with its likelihood Eq. (9), the posterior of $(\mu_i, \sigma_i^2)$ differs from the PMB prior as follows:

$$P(\mu_i | y_i, \sigma_i^2)_{\text{rfb}} \propto \exp\left\{-\frac{1}{2\sigma_{i,\text{rfb}}^2} \sum_{j:x_{ij}>0} (\ln y_{ij} - \mu_i)^2\right\} \tag{16}$$

$$P(\sigma_i^2 | y_i)_{\text{rfb}} \quad \propto (\sigma_i^2)^{-\frac{n_{i1}}{2}} \sqrt{1 + (2\sigma_i^2)^{-1}} \exp\left\{-\frac{(n_{i1}-1)\hat{\sigma}_i^2}{2\sigma_i^2}\right\}. \tag{17}$$

Moreover, it can be similarly denoted as $d_{i,\text{rfb}}^{(post)} | y_i \sim \text{beta}(n_{i0} + 1/2, n_{i1} + 1/2)$, $\mu_{i,\text{rfb}}^{(post)} | y_i \sim N(\hat{\mu}_{i,\text{rfb}}, \sigma_{i,\text{rfb}}^{2(post)})$ and $\sigma_{i,\text{rfb}}^{2(post)} \propto (\sigma_i^2)^{-\frac{n_{i1}}{2}} \sqrt{1 + (2\sigma_i^2)^{-1}} \exp\left\{-\frac{(n_{i1}-1)\hat{\sigma}_i^2}{2\sigma_i^2}\right\}$, respectively. The posterior of $\lambda_{ik}$ is $\lambda_{ik,\text{rfb}}^{(post)} = T_{i,\text{rfb}}^{(post)} - T_{k,\text{rfb}}^{(post)}$, where $T_{i,\text{rfb}}^{(post)} \approx \ln d_{i,\text{rfb}}^{(post)} + 2(\mu_i^{(post)} + \sigma_{i,\text{rfb}}^{2(post)})$ and $T_{k,\text{rfb}}^{(post)} \approx \ln d_{k,\text{rfb}}^{(post)} + 2(\mu_{k,\text{rfb}}^{(post)} + \sigma_{k,\text{rfb}}^{2(post)})$. According to *Ganesh (2009)*, the $100(1-\alpha)\%$ Bayesian-based SCI with the RB prior for $\lambda_{ik}$ is

$$[L_{\lambda_{ik}}, U_{\lambda_{ik}}]_{\text{rfb}} = \left[\lambda_{ik,\text{rfb}}^{(post)} \mp v_\alpha^{\text{rfb}}\right], \tag{18}$$

where $v_\alpha^{\text{rfb}}$ stands for the $(1-\alpha)^{th}$ percentile of the distribution of $V^{\text{rfb}} = \max_h\left\{\lambda_{ik,\text{rfb}}^{(post)}\right\} - \min_h\left\{\lambda_{ik,\text{rfb}}^{(post)}\right\}$.

### The GPQ approach

Motivated by *Wu & Hsieh (2014)*, the GPQ of $d_i$ is formulated using the arcsin square-root transformation of the variance. Moreover, the GPQs for $(\mu_i, \sigma_i^2)$ are also obtained from transformation of the normal approximation by using the central limit theorem (*Tian, 2005*; *Hasan & Krishnamoorthy, 2017*). The GPQ for $T_i$ can be written as

$$G_{T_i} = \ln\left[1 - \sin^2\left\{\sin^{-1}\sqrt{\hat{d}_i} - \frac{R_i}{2\sqrt{n_i}}\right\}\right] + 2\left[\hat{\mu}_i - S_i\sqrt{\frac{G_{\sigma_i^2}}{n_{i1}}} + G_{\sigma_i^2}\right], \tag{19}$$

where $G_{\sigma_i^2} = (n_{i1} - 1)\hat{\sigma}_i^2 / U_i$. The random variables $R_i = 2\sqrt{n_i}\left(\sin^{-1}\sqrt{\hat{d}_i} - \sin^{-1}\sqrt{d_i}\right)$, $S_i = (\hat{\mu}_i - \mu_i)/(\sigma_i^2/n_{i1})$ and $U_i = (n_{i1} - 1)\hat{\sigma}_i^2 / \sigma_i^2$ are independent from standard normal, normal and $\chi_{n_{i1}-1}^2$ distributions, respectively. Thus, the corresponding GPQ of $\lambda_{ik}$ can be expressed as

$$G_{\lambda_{ik}} = G_{T_i} - G_{T_k}. \tag{20}$$

Similarly, $G_{T_k} = \ln(1 - G_{d_k}) + G_{2\mu_k} + G_{2\sigma_k^2}$ denotes the GPQ of $T_k$; $G_{d_k} = \sin^2\left\{\sin^{-1}\sqrt{\hat{d}_k} - \left[R_k(2\sqrt{n_k})^{-1}\right]\right\}$, $G_{2\mu_k} = 2\left(\hat{\mu}_k - S_k\sqrt{G_{\sigma_k^2}/n_{k1}}\right)$, and $G_{2\sigma_k^2} = 2(n_{k1} - 1)\hat{\sigma}_k^2 / U_k$. Therefore, the $100(1-\alpha)\%$ SCI for $\lambda_{jk}$ based on the GPQ approach is given by

$$[L_{\lambda_{ik}}, U_{\lambda_{ik}}]_{\text{gpq}} = \left[\hat{\lambda}_{ik} \mp q_\alpha^{\text{GPQ}}\sqrt{\widehat{Var}(\hat{\lambda}_{ik})}\right], \tag{21}$$

where $q_\alpha^{\text{GPQ}}$ denotes the $(1-\alpha)^{th}$ percentile of the $Q^{\text{GPQ}}$ distribution; the $Q^{\text{GPQ}}$ is derived as

$$Q^{\text{GPQ}} = \max_{j \neq l} \left| \left\{ \hat{\lambda}_{ik} - G_{\lambda_{ik}}(Y, Y^*, d, \mu, \sigma^2) \right\} / \sqrt{\widehat{Var}(\hat{\lambda}_{ik})} \right|. \tag{22}$$

In agreement with *Hannig et al. (2006)*, *Kharrati-Kopaei & Eftekhar (2017)*, the asymptotic coverage probability of the SCI for $\lambda_{ik}$ based on the GPQ is slightly modified from that in *Maneerat, Niwitpong & Niwitpong (2021b)* (the proof of Theorem 1 in the Appendix).

**Theorem 1** *Let* $Y_i = (Y_{i1}, Y_{i2}, \ldots, Y_{in_i}) \overset{iid}{\sim} ZILN(d_i, \mu_i, \sigma_i^2)$. *For* $Y_i = 0$, $n_{i0}$ *is binomially distributed with the proportion of zero inflation* $d_i = E(n_{i0}/n_i)$. *For* $Y_i > 0$, $\ln Y_i$ *is log-normally distributed with mean* $\mu_i = E(\ln Y_i)$ *and variance* $\sigma_i^2 = Var(\ln Y_i)$. *Moreover, let* $\lambda_{ik} = T_i/T_k$; $T_i \approx \ln d_i + 2(\mu_i + \sigma_i^2)$ *from group* $i$ *be the log-transformed variance of ZILN. Given* $y_i = (y_{i1}, y_{i2}, \ldots, y_{in_i})$, *let* $\widehat{Var}(\hat{\lambda}_{ik})$ *be an approximated variance of* $\hat{\lambda}_{ik} = \hat{T}_i/\hat{T}_k$, *where* $(\hat{T}_i, \hat{T}_k)$ *are the estimates of* $(T_i, T_k)$. *Suppose that* $n_i/n \to \varphi_i \in (0,1)$ *as* $n = \sum_{i=1}^{h} n_i \to \infty$, *thus it follows that the asymptotically coverage probability of* $100(1-\alpha)\%$ *SCI for* $\lambda_{jk}$ *based the GPQ approach is given by*

$$P\left( \lambda_{jk} \in \left[ \hat{\lambda}_{ik} \mp q_\alpha^{\text{GPQ}} \sqrt{\widehat{Var}(\hat{\lambda}_{ik})} \right] \right) \to 1 - \alpha \tag{23}$$

*for* $\forall i \neq k$ *and* $i, k = 1, \ldots, h$.

### The PB approach

Here, we assume that the data come from a known distribution with unknown parameters that are estimated by using samples stimulated from the estimated distribution. In the present study, the PB approach is adjusted to suit our particular situation. Let $\hat{d}_i^*$, $\hat{\mu}_i^*$ and $\hat{\sigma}_i^{2*}$ be the observed values of $\hat{d}_i$, $\hat{\mu}_i$, and $\hat{\sigma}_i^2$ representing the estimated values of parameters $d_i$, $\mu_i$, and $\sigma_i^2$, respectively. Thus, we can obtain the empirical distribution of $T$ based on the PB approach. In accordance with *Sadooghi-Alvandi & Malekzadeh (2014)*, the respective sampling distributions of $(\hat{d}_i^*, \hat{\mu}_i^*, \hat{\sigma}_i^{2*})$ are

$$\hat{d}_i^{(pboot)} \sim \text{beta}(n_{i0}^* + 1/2, n_{i1}^* + 1/2) \tag{24}$$

$$\hat{\mu}_i^{(pboot)} = \hat{\mu}_i^* + D_i \sqrt{\frac{\hat{\sigma}_i^{2*}}{n_{i1}^*}} \tag{25}$$

$$\hat{\sigma}_i^{2(pboot)} = \frac{\hat{\sigma}_i^{2*} U}{n_{i1}^* - 1}, \tag{26}$$

where $D_i = [\hat{\mu}_i^{(pboot)} - \hat{\mu}_i^*] / \sqrt{\hat{\sigma}_j^{2*}/n_{i1}^*} \sim N(0,1)$ and $U_j = [n_{i1}^* - 1]\hat{\sigma}_i^{2(pboot)} / \hat{\sigma}_i^{2*} \sim \chi_{n_{i1}^*-1}^2$ are independent random variables with standard normal and Chi-square distributions, respectively. The PB variable-based pivotal quantity is expressed as

$$M^{\text{PB}} = \left| \left( \hat{\lambda}_{ik}^{pboot} - \hat{\lambda}_{ik}^* \right) / \sqrt{\widehat{Var}(\hat{\lambda}_{ik}^*)} \right|, \tag{27}$$

where $\hat{\lambda}_{ik}^{(pboot)} = \hat{T}_i^{(pboot)} - \hat{T}_k^{(pboot)}$ and $\hat{\lambda}_{ik}^* = \hat{T}_i^* - \hat{T}_k^*$. By replacing observed values $(\hat{d}_i^* \hat{\mu}_i^*, \hat{\sigma}_i^{2*})$ from the samples, we respectively obtain

$$\hat{\lambda}_{ik}^* = \ln\left(\frac{\hat{d}_i^*}{\hat{d}_k^*}\right) + 2\left[(\hat{\mu}_i^* - \hat{\mu}_k^*) + (\hat{\sigma}_i^{2*} - \hat{\sigma}_k^{2*})\right] \tag{28}$$

$$\hat{\lambda}_{ik}^{(pboot)} = \ln\left(\frac{\hat{d}_i^{(pboot)}}{\hat{d}_k^{(pboot)}}\right) + 2\left[(\hat{\mu}_i^{(pboot)} - \hat{\mu}_k^{(pboot)}) + (\hat{\sigma}_i^{2(pboot)} - \hat{\sigma}_k^{2(pboot)})\right] \tag{29}$$

$$\hat{Var}(\hat{\lambda}_{ik}^*) = \frac{\hat{d}_i^*}{n_i \hat{d}_i^*} + \frac{\hat{d}_i^*}{n_k \hat{d}_k^*} + 4\left(\frac{\hat{\sigma}_i^{2*}}{n_{i1}^*} + \frac{2\hat{\sigma}_i^{2*}}{n_{i1}^* - 1} + \frac{\hat{\sigma}_k^{2*}}{n_{k1}^*} + \frac{2\hat{\sigma}_k^{2*}}{n_{k1}^* - 1}\right), \tag{30}$$

where $\hat{d}_i^* = 1 - \hat{d}_i^*$ and $n_{i1}^* = n_i \hat{d}_i^*$. Hence, the $100(1-\alpha)\%$ SCI for $\lambda_{ik}$ based on the PB approach is

$$[L_{\lambda_{ik}}, U_{\lambda_{ik}}]_{(PB)} = \left[\hat{\lambda}_{ik} \mp M_\alpha^{PB}\sqrt{\hat{Var}(\hat{\lambda}_{ik})}\right], \tag{31}$$

where $m_\alpha^{PB}$ is the $(1-\alpha)^{th}$ percentile of the distribution of $M^{PB}$. Theorem 2 shows the asymptotic coverage probability of the $100(1-\alpha)\%$ SCI for $\lambda_{ik}$ based on the PB approach (see the proof in the Appendix ).

**Theorem 2** *Suppose that* $Y_i = (Y_{i1}, Y_{i2}, \ldots, Y_{in_i})$ *comprise an iid random vector from a ZILN model based on* $n_i$ *observations from population group* $i$. *Let* $\hat{\lambda}_{ik} = \hat{T}_i - \hat{T}_k$ *be the estimate of* $\lambda_{ik}$, *where* $\hat{T}_i$ *and* $\hat{T}_k$ *are the approximately log-transformed variances of* $\hat{T}_i$ *and* $\hat{T}_k$ *from the population groups* $i$th *and* $k$th, *respectively. Hence,*

$$P\left[\lambda_{ik} \in \left(\hat{\lambda}_{ik} \mp M_\alpha^{PB}\sqrt{\hat{Var}(\hat{\lambda}_{ik})}\right)\right] \to 1 - \alpha, \tag{32}$$

*where* $\hat{Var}(\hat{\lambda}_{ik})$ *is the estimated variance of* $\hat{\lambda}_{ik}$; $\forall i \neq k$ *and* $i, k = 1, 2, .., h$.

## SIMULATION STUDIES AND RESULTS

Simulation studies were conducted to assess the performances of the SCIs based Bayesian, GPQ, and PB approaches for all pairwise ratios of variances of several ZILN distributions: Bayesian SCIs based on PMB and RB priors (*Maneerat, Niwitpong & Niwitpong, 2020*), the GPQ-based SCI (*Wu & Hsieh, 2014*), and the PB-based SCI (*Sadooghi-Alvandi & Malekzadeh, 2014*; *Li, Song & Shi, 2015*; *Kharrati-Kopaei & Eftekhar, 2017*). CRs, LERs, UERs, and AWs of the SCIs were determined when the population group size($h$) were fixed at 3 and 5; the optimal values of CR, LER, UER, and AW are 95%, 5%, 5% and 0, respectively, which were used to judge the best-performing SCI. Critical values $v_\alpha^{pmb}$, $v_\alpha^{rb}$, $q_\alpha^{GPQ}$ and $m_\alpha^{PB}$ for the Bayesian SCIs based on PMB and RB priors, GPQ and PB, respectively, were also assessed. Throughout the simulation studies, the simulation procedure to estimate the CRs, LERs, and UERs was as follows:

(i)   Generate random samples $Y_i = (Y_{i1}, Y_{i2}, \ldots, Y_{in_i})$ from $ZILN(d_i, \mu_i, \sigma_i^2)$, and compute $\hat{d}_i, \hat{\mu}_i, \hat{\sigma}_i^2$; $i = 1, 2, \ldots, h$ from the samples.

(ii)   Compute the critical values for each method using 2500 Monte Carlo simulations.

(iii)   Apply the SCIs based on Bayesian-based PMB and RB priors, GPQ, and PB approaches given in Eqs. (14), (18), (21) and (31), respectively, and record whether or not the values of $(\lambda_{ik}; i \neq k)$ fall within their corresponding confidence intervals.

(iv)  Repeat steps (i)-(iii) $M = 5000$ times.

(v)  For each method: obtain the number of times that all $(\lambda_{ik}; i \neq k)$ are in their corresponding SCIs to estimated the CR.

(vi)  Obtain the number of times that all $(\lambda_{ik}; i \neq k)$ is less than or greater than their corresponding SCIs to estimate the LER and UER, respectively.

For the three-group comparison, the following parameter combinations were used: large variances $(\sigma_1^2, \sigma_2^2, \sigma_3^2) = (3, 5, 7)$; small $(30, 30, 30)$, moderate $(50, 50, 50)$, large $[(100, 100, 100)$ and $(100, 100, 200)]$, small-to-large $(30, 50, 100)$ and medium-to-large $(50, 100, 200)$ sample sizes; and zero-inflation percentages of $(10, 20, 30)$, $(10, 30, 50)$ and $(30, 50, 50)$. For the five-group comparison, the following parameter combinations were used: large variances $(\sigma_1^2, \sigma_2^2, \sigma_3^2, \sigma_4^2, \sigma_5^2) = (1, 1, 2, 2, 3)$; small-to-large $(30, 50, 50, 100, 200)$, medium-to-large $(50, 50, 50, )(100, 100)$, and large $(70, 100, 100, 200, 200)$ sample sizes; and zero-inflation percentages of $(10, 10, 20, 20, 20)$, $(20, 20, 30, 30, 50)$ and $(50, 50, 50, 70, 70)$. The results are reported in Table 1.

For $h = 3$ with large variance, Table 1 and Fig. 1 reveal that all of the methods provided CR performances close to and greater than the nominal confidence level (95%). Meanwhile, the SCIs based on the Bayesian approach based on the PMB prior and GPQ maintained a good balance between LER and UER. Importantly, the AW of PB was narrower than the other methods for small sample sizes, while those of the Bayesian approach based on the PMB prior were slightly narrower than the others for the other sample sizes. When a group comparison was $h = 5$ (Table 1 and Fig. 2), the PB approach provided the best CRs and narrowest AWs for all scenarios tested.

## AN EMPIRICAL APPLICATION OF THE FOUR METHODS TO DAILY PRECIPITATION DATA

Daily precipitation records comprise publicly available data from the Thailand Meteorology Department (*Department, 2021*). Flash floods, landslides, and windstorms caused by heavy rainfall occurred in the four provinces in the lower southern area of Thailand: Songkhla, Yala, Narathiwat, and Pattani during January 2021, as reported by Thailand's Department of Disaster Prevention and Mitigation (*Thailand, 2021*). According to automatic weather system (*Department, 2021*), Songkhla has two weather stations in the Songkhla and Sadao districts, which means that we could simultaneously estimate variations in precipitation at five weather stations.

Daily precipitation data from December 2020 to January 2021 (Table 2) were used in the analysis. Figure 3 shows histogram along with normal quantile–quantile (Q-Q), cumulative density function (CDF) and probability-probability (P-P) plots. Furthermore, the Akaike information criterion (AIC) and Bayesian information criterion (BIC) values of five models: normal, logistic, lognormal, exponential, and Cauchy applied to fitting the non-zero precipitation data were compared to check the appropriateness of each model for fitting the data (Table 3). The AIC and BIC results for the lognormal model

Maneerat and Niwitpong (2021), *PeerJ*, DOI 10.7717/peerj.12659

**Table 1   Performance measures of SCIs-based different approaches.**

| $n_i$ | $d_i$ (%) | B-PMB | | | B-RB | | | GPQ | | | PB | | | AW | | | |
|---|---|---|---|---|---|---|---|---|---|---|---|---|---|---|---|---|---|
| | | LER | CR | UER | LER | CR | UER | LER | CR | UER | LER | CR | UER | B-PMB | B-RB | GPQ | PB |
| 3 sample groups and $(\sigma_1^2, \sigma_2^2, \sigma_3^2) = (3,5,7)$ | | | | | | | | | | | | | | | | | |
| $(30_3)$ | (10,20,30) | 1.993 | 97.973 | 0.033 | 1.307 | 98.693 | 0.000 | 0.707 | 99.293 | 0.000 | 2.460 | 97.540 | 0.000 | 22.961 | 25.493 | 27.764 | **22.942** |
| | (10,30,50) | 1.880 | 98.113 | 0.007 | 1.200 | 98.800 | 0.000 | 0.967 | 99.033 | 0.000 | 2.900 | 97.100 | 0.000 | 28.702 | 33.323 | 33.300 | **27.254** |
| | (30,50,50) | 1.120 | 98.873 | 0.007 | 0.427 | 99.573 | 0.000 | 0.620 | 99.380 | 0.000 | 2.520 | 97.480 | 0.000 | 30.764 | 35.737 | 36.813 | **29.113** |
| $(50_3)$ | (10,20,30) | 2.833 | 96.800 | 0.367 | 2.300 | 97.567 | 0.133 | 1.107 | 98.887 | 0.007 | 2.347 | 97.627 | 0.027 | **15.521** | 16.544 | 19.078 | 16.893 |
| | (10,30,50) | 2.887 | 97.027 | 0.087 | 2.173 | 97.800 | 0.027 | 1.253 | 98.747 | 0.000 | 2.607 | 97.393 | 0.000 | **18.848** | 20.654 | 22.403 | 19.733 |
| | (30,50,50) | 2.087 | 97.840 | 0.073 | 1.413 | 98.567 | 0.020 | 0.973 | 99.027 | 0.000 | 2.320 | 97.673 | 0.007 | **20.104** | 21.996 | 24.567 | 21.096 |
| $(100_3)$ | (10,20,30) | 3.480 | 95.140 | 1.380 | 3.200 | 95.693 | 1.107 | 1.273 | 98.527 | 0.200 | 1.960 | 97.767 | 0.273 | **10.015** | 10.325 | 12.448 | 11.681 |
| | (10,30,50) | 3.660 | 95.627 | 0.713 | 3.200 | 96.420 | 0.380 | 1.427 | 98.540 | 0.033 | 2.087 | 97.833 | 0.080 | **11.866** | 12.410 | 14.327 | 13.422 |
| | (30,50,50) | 3.220 | 96.040 | 0.740 | 2.780 | 96.747 | 0.473 | 1.167 | 98.813 | 0.020 | 2.073 | 97.853 | 0.073 | **12.389** | 12.948 | 15.408 | 14.202 |
| (30,50,100) | (10,20,30) | 1.787 | 96.753 | 1.460 | 1.367 | 97.453 | 1.180 | 0.380 | 99.480 | 0.140 | 1.127 | 98.467 | 0.407 | **12.846** | 13.402 | 16.552 | 14.152 |
| | (10,30,50) | 1.853 | 97.127 | 1.020 | 1.387 | 97.993 | 0.620 | 0.420 | 99.553 | 0.027 | 1.420 | 98.353 | 0.227 | **14.348** | 15.042 | 18.368 | 15.604 |
| | (30,50,50) | 1.013 | 97.947 | 1.040 | 0.547 | 98.687 | 0.767 | 0.260 | 99.653 | 0.087 | 1.053 | 98.627 | 0.320 | **16.343** | 17.452 | 20.826 | 17.181 |
| (50,100,200) | (10,20,30) | 2.580 | 94.773 | 2.647 | 2.247 | 95.293 | 2.460 | 0.467 | 99.047 | 0.487 | 0.847 | 98.307 | 0.847 | **8.637** | 8.822 | 11.261 | 10.230 |
| | (10,30,50) | 2.847 | 95.073 | 2.080 | 2.593 | 95.560 | 1.847 | 0.667 | 99.093 | 0.240 | 1.313 | 98.140 | 0.547 | **9.522** | 9.725 | 12.334 | 11.166 |
| | (30,50,50) | 2.173 | 95.880 | 1.947 | 1.793 | 96.533 | 1.673 | 0.380 | 99.380 | 0.240 | 1.020 | 98.460 | 0.520 | **10.618** | 10.939 | 13.751 | 12.189 |
| $(100_2, 200)$ | (10,20,30) | 3.253 | 94.213 | 2.533 | 2.953 | 94.693 | 2.353 | 0.967 | 98.673 | 0.360 | 1.507 | 97.920 | 0.573 | **7.952** | 8.090 | 10.266 | 9.647 |
| | (10,30,50) | 2.940 | 95.013 | 2.047 | 2.620 | 95.533 | 1.847 | 0.980 | 98.793 | 0.227 | 1.460 | 98.127 | 0.413 | **8.985** | 9.184 | 11.489 | 10.773 |
| | (30,50,50) | 2.567 | 95.387 | 2.047 | 2.227 | 96.007 | 1.767 | 0.900 | 98.893 | 0.207 | 1.547 | 98.047 | 0.407 | **9.888** | 10.197 | 12.666 | 11.709 |
| 5 sample groups and $(\sigma_1^2, \sigma_2^2, \sigma_3^2, \sigma_4^2, \sigma_5^2) = (1,1,2,2,3)$ | | | | | | | | | | | | | | | | | |
| $(30, 50_2, 100, 200)$ | (10,10,20,20,20) | 0.326 | 99.504 | 0.170 | 0.232 | 99.626 | 0.142 | 0.344 | 99.568 | 0.088 | 0.756 | 99.002 | 0.242 | 6.224 | 6.471 | 6.310 | **5.600** |
| | (20,20,30,30,50) | 0.244 | 99.620 | 0.136 | 0.154 | 99.754 | 0.092 | 0.244 | 99.694 | 0.062 | 0.666 | 99.164 | 0.170 | 6.952 | 7.250 | 7.067 | **6.201** |
| | (20,30,50,50,70) | 0.154 | 99.738 | 0.108 | 0.092 | 99.828 | 0.080 | 0.322 | 99.642 | 0.036 | 0.788 | 99.084 | 0.128 | 8.510 | 8.971 | 8.513 | **7.369** |
| | (50,50,50,70,70) | 0.062 | 99.882 | 0.056 | 0.026 | 99.942 | 0.032 | 0.116 | 99.872 | 0.012 | 0.426 | 99.490 | 0.084 | 9.572 | 10.226 | 9.861 | **8.223** |
| $(50_3, 100_2)$ | (10,10,20,20,20) | 0.398 | 99.504 | 0.098 | 0.338 | 99.582 | 0.080 | 0.558 | 99.414 | 0.028 | 1.122 | 98.788 | 0.090 | 6.614 | 6.826 | 6.557 | **5.914** |
| | (20,20,30,30,50) | 0.392 | 99.512 | 0.096 | 0.312 | 99.618 | 0.070 | 0.526 | 99.448 | 0.026 | 1.092 | 98.810 | 0.098 | 7.791 | 8.100 | 7.567 | **6.768** |
| | (20,30,50,50,70) | 0.358 | 99.618 | 0.024 | 0.244 | 99.748 | 0.008 | 0.582 | 99.398 | 0.020 | 1.196 | 98.754 | 0.050 | 10.067 | 10.737 | 9.488 | **8.354** |
| | (50,50,50,70,70) | 0.204 | 99.766 | 0.030 | 0.136 | 99.850 | 0.014 | 0.254 | 99.746 | 0.000 | 0.822 | 99.166 | 0.012 | 10.687 | 11.352 | 10.571 | **9.039** |
| $(70, 100_2, 200_2)$ | (10,10,20,20,20) | 0.784 | 99.038 | 0.178 | 0.710 | 99.140 | 0.150 | 0.810 | 99.120 | 0.080 | 1.232 | 98.640 | 0.128 | 4.499 | 4.565 | 4.507 | **4.237** |
| | (20,20,30,30,50) | 0.666 | 99.174 | 0.160 | 0.580 | 99.280 | 0.140 | 0.620 | 99.310 | 0.070 | 1.058 | 98.826 | 0.116 | 5.218 | 5.321 | 5.116 | **4.783** |
| | (20,30,50,50,70) | 0.620 | 99.290 | 0.090 | 0.550 | 99.380 | 0.060 | 0.750 | 99.200 | 0.060 | 1.158 | 98.744 | 0.098 | 6.546 | 6.743 | 6.202 | **5.753** |
| | (50,50,50,70,70) | 0.374 | 99.548 | 0.078 | 0.310 | 99.630 | 0.060 | 0.370 | 99.600 | 0.030 | 0.680 | 99.258 | 0.062 | 6.938 | 7.139 | 6.892 | **6.249** |

**Notes.**

Note: $(100_3, 200_2) = (100, 100, 100, 200, 200)$. Bold denotes the best-performing method.

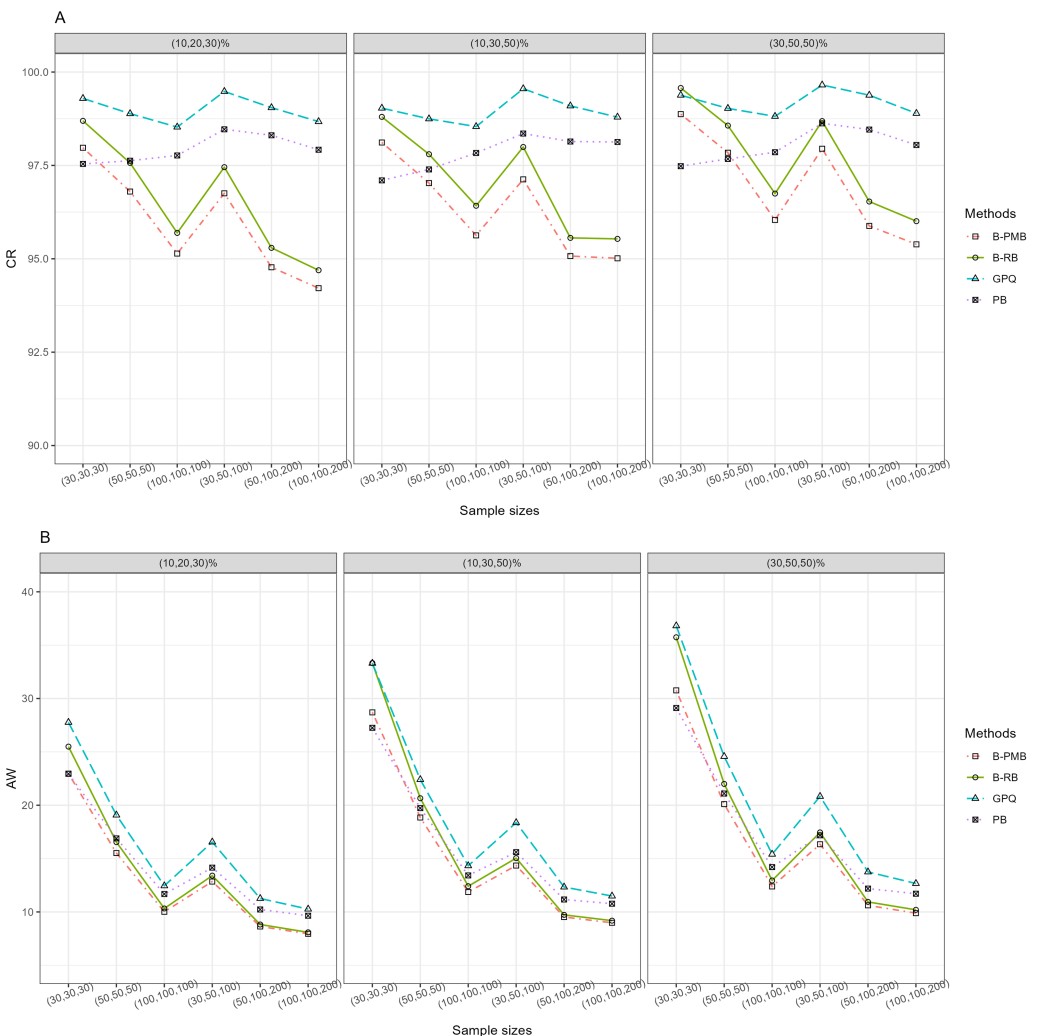

**Figure 1** The CR and AW performance measures for three sample groups: (A) CR (B) AW.

were the lowest, and thus it was the most efficient. The data from all of the stations were zero-inflated, thereby verifying that they follow the assumptions for ZILN.

The results in Table 4 reveals that since variance $\sigma_i^2$ was greater than the mean $\mu_i$, quite large precipitation variations were required in the present study. For applying data of daily precipitation to measure the efficacy of the four methods, the 95% SCIs-based Bayesian, GPQ and PB approaches for all pairwise precipitation datasets from the five weather stations cover their point estimates (Table 5). In a agree with the simulation results for $n_1 = n_2 = n_3 = 50$ and $n_4 = n_5 = 100$, the PB approach provided the best SCI performance for ratio of variances of several ZILN models. This can be interpreted as Narathiwat has the highest variation in precipitation, followed by Yala. These results are in line with the Asia Disaster Monitoring and Response System (*Thailand, 2021*), which reported that both areas were affected by flooding and landslides damaging 22,308 households in Narathiwat

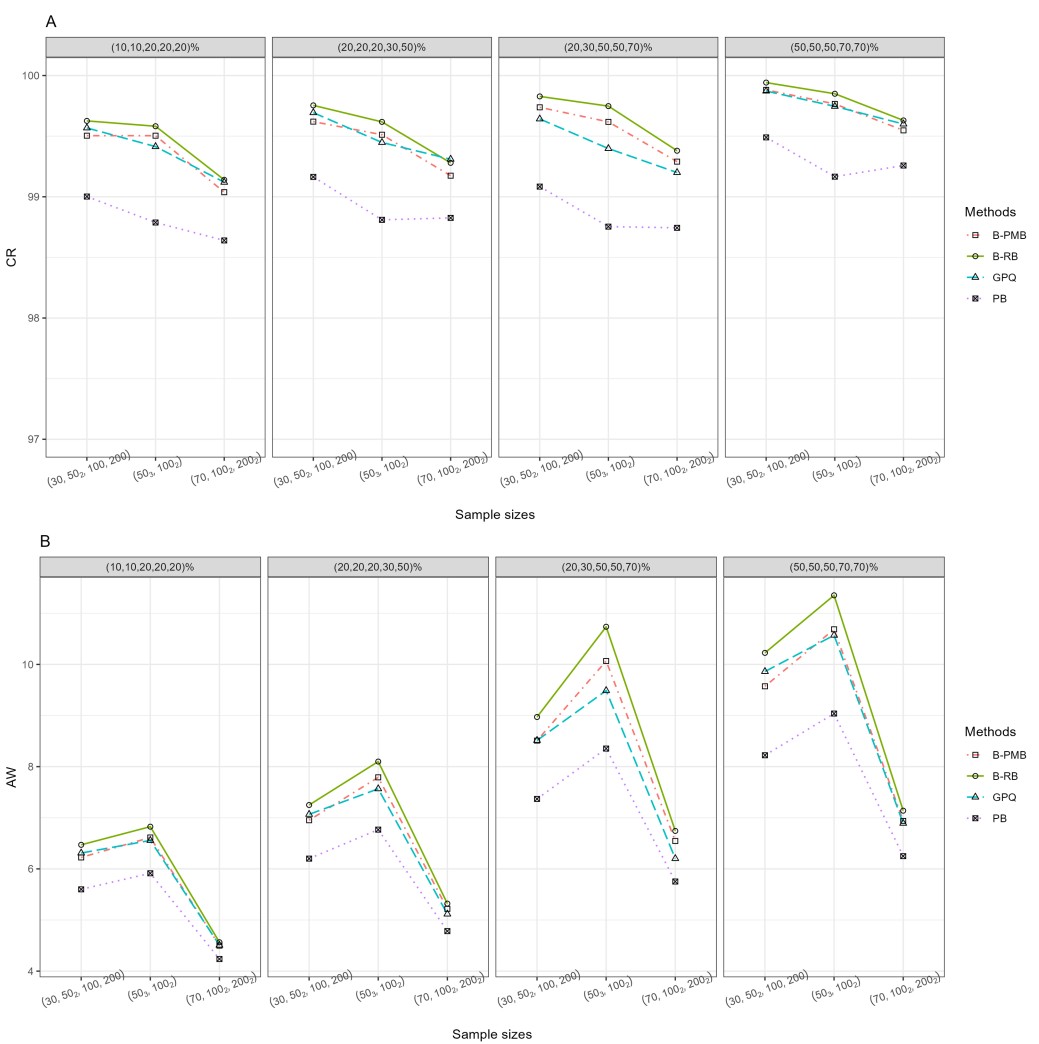

**Figure 2** The CR and AW performance measures for five sample groups: (A) CR (B) AW.

and 12,082 households in Yala during the time period covered by the data used in the study.

## DISCUSSION

From the above numerical results, it can be seen that the SCIs based on PB and the Bayesian approach based on the PMB prior dealt with large variations in the data better than the other approaches. The PB-based SCI has some strong points for small sample sizes due to random samples being obtained *via* bootstrap resampling. Furthermore, the performance of the Bayesian SCI based on the PMB prior declined as the number of populations increased and the sample size decreased. Although, the GPQ method provided appropriate CRs, its AWs were wider than the other methods, possibly because the GPQ of $d_i$ is limited for cases with unequal zero-inflated percentages. Since it has performed quite well for

**Table 2  Daily precipitation data in five stations of southern Thailand.**

| Dates | Weather stations: December 2020 | | | | | Dates | Weather stations: January 2021 | | | | |
|---|---|---|---|---|---|---|---|---|---|---|---|
| | Shongklha | Songkhla-based Sadao district | Yala | Narathiwat | Pattani | | Shongklha | Songkhla-based Sadao district | Yala | Narathiwat | Pattani |
| 1 | 160.0 | 56.4 | 46.4 | 38.6 | 82.0 | 1 | 0.8 | 4.2 | 6.6 | 31.2 | 0.8 |
| 2 | 14.6 | 85.8 | 46.6 | 70.0 | 0.0 | 2 | 1.4 | 8.2 | 5.6 | 6.4 | 2.0 |
| 3 | 20.8 | 4.2 | 55.8 | 74.2 | 0.0 | 3 | 2.6 | 42.6 | 49.6 | 38.6 | 49.8 |
| 4 | 8.8 | 0.2 | 27.0 | 0.4 | 7.2 | 4 | 21.4 | 8.4 | 28.6 | 10.4 | 4.4 |
| 5 | 0.0 | 0.0 | 0.2 | 0.0 | 0.0 | 5 | 9.2 | 70.2 | 137.8 | 62.8 | 49.0 |
| 6 | 0.0 | 0.0 | 0.0 | 0.0 | 0.2 | 6 | 0.2 | 2.8 | 84.8 | 13.2 | 0.2 |
| 7 | 0.0 | 0.0 | 0.0 | 0.0 | 0.0 | 7 | 0.0 | 0.0 | 1.8 | 9.2 | 2.8 |
| 8 | 0.2 | 0.0 | 1.6 | 0.0 | 0.0 | 8 | 0.4 | 0.0 | 0.4 | 0.0 | 1.2 |
| 9 | 52.0 | 0.0 | 0.0 | 0.0 | 0.0 | 9 | 0.8 | 0.0 | 0.0 | 1.4 | 0.0 |
| 10 | 39.4 | 0.0 | 0.0 | 0.8 | 3.6 | 10 | 29.0 | 15.6 | 2.8 | 12.6 | 22.8 |
| 11 | 0.6 | 0.0 | 2.8 | 9.2 | 9.8 | 11 | 23.0 | 0.6 | 0.2 | 0.2 | 0.0 |
| 12 | 12.2 | 4.2 | 17.2 | 0.0 | 8.0 | 12 | 5.0 | 0.2 | 0.6 | 3.6 | 1.2 |
| 13 | 5.4 | 37.2 | 2.0 | 8.2 | 12.8 | 13 | 0.0 | 0.0 | 2.4 | 3.0 | 1.0 |
| 14 | 9.4 | 0.0 | 0.0 | 0.0 | 3.4 | 14 | 5.4 | 0.0 | 0.0 | 0.0 | 0.0 |
| 15 | 7.0 | 2.4 | 12.4 | 78.4 | 7.2 | 15 | 1.8 | 0.0 | 0.0 | 0.0 | 0.0 |
| 16 | 19.2 | 25.6 | 43.8 | 43.0 | 62.8 | 16 | 0.8 | 0.0 | 0.0 | 0.0 | 0.0 |
| 17 | 84.4 | 97.4 | 126.4 | 162.0 | 164.8 | 17 | 0.0 | 0.0 | 0.0 | 0.0 | 0.0 |
| 18 | 97.2 | 9.2 | 113.8 | 141.2 | 46.4 | 18 | 0.0 | 0.0 | 0.0 | 1.2 | 0.0 |
| 19 | 92.0 | 19.2 | 39.8 | 43.6 | 26.2 | 19 | 0.0 | 0.0 | 0.0 | 0.0 | 0.0 |
| 20 | 19.8 | 7.2 | 27.8 | 20.4 | 7.0 | 20 | 0.0 | 0.0 | 0.0 | 0.0 | 0.0 |
| 21 | 5.4 | 0.4 | 0.0 | 0.2 | 3.4 | 21 | 0.0 | 0.0 | 0.0 | 0.0 | 0.0 |
| 22 | 0.0 | 0.0 | 1.2 | 1.0 | 3.4 | 22 | 0.0 | 0.0 | 0.0 | 0.0 | 0.0 |
| 23 | 23.8 | 0.0 | 31.0 | 61.4 | 12.6 | 23 | 0.0 | 0.0 | 0.0 | 0.0 | 0.0 |
| 24 | 23.4 | 0.0 | 19.6 | 6.6 | 0.0 | 24 | 0.0 | 0.0 | 2.2 | 0.0 | 0.0 |
| 25 | 2.2 | 0.0 | 46.6 | 39.8 | 6.8 | 25 | 0.0 | 0.0 | 0.0 | 0.0 | 0.0 |
| 26 | 1.0 | 10.0 | 27.6 | 84.0 | 2.8 | 26 | 0.0 | 0.0 | 0.0 | 0.0 | 0.0 |
| 27 | 0.0 | 0.0 | 1.0 | 0.0 | 0.2 | 27 | 0.0 | 0.0 | 2.0 | 0.2 | 0.0 |
| 28 | 0.0 | 0.0 | 0.0 | 0.0 | 0.0 | 28 | 0.0 | 0.0 | 0.0 | 0.0 | 0.0 |
| 29 | 0.0 | 0.0 | 0.0 | 0.0 | 0.0 | 29 | 0.0 | 0.0 | 0.0 | 0.0 | 0.0 |
| 30 | 0.0 | 0.0 | 0.0 | 0.0 | 0.0 | 30 | 4.4 | 0.0 | 0.0 | 0.0 | 0.0 |
| 31 | 6.2 | 0.4 | 11.2 | 89.2 | 3.2 | 31 | 9.6 | 2.0 | 3.0 | 0.4 | 1.6 |

**Notes.**

Source: Thailand Meteorological Department Automatic Weather System.
http://www.aws-observation.tmd.go.th/web/climate/climate_past.asp.

one population group especially (*Wu & Hsieh, 2014*; *Maneerat, Niwitpong & Niwitpong, 2021a*). Further research could be conducted to explore subjective or prior beliefs about parameters when using the Bayesian approach for parameter estimation

# CONCLUSIONS

SCIs for the comparison of the variance ratios among several ZILN models were formulated by applying Bayesian approaches based on the PMB and RB priors, along with the GPQ and PB approaches. In practice, the daily precipitation data for each of the weather

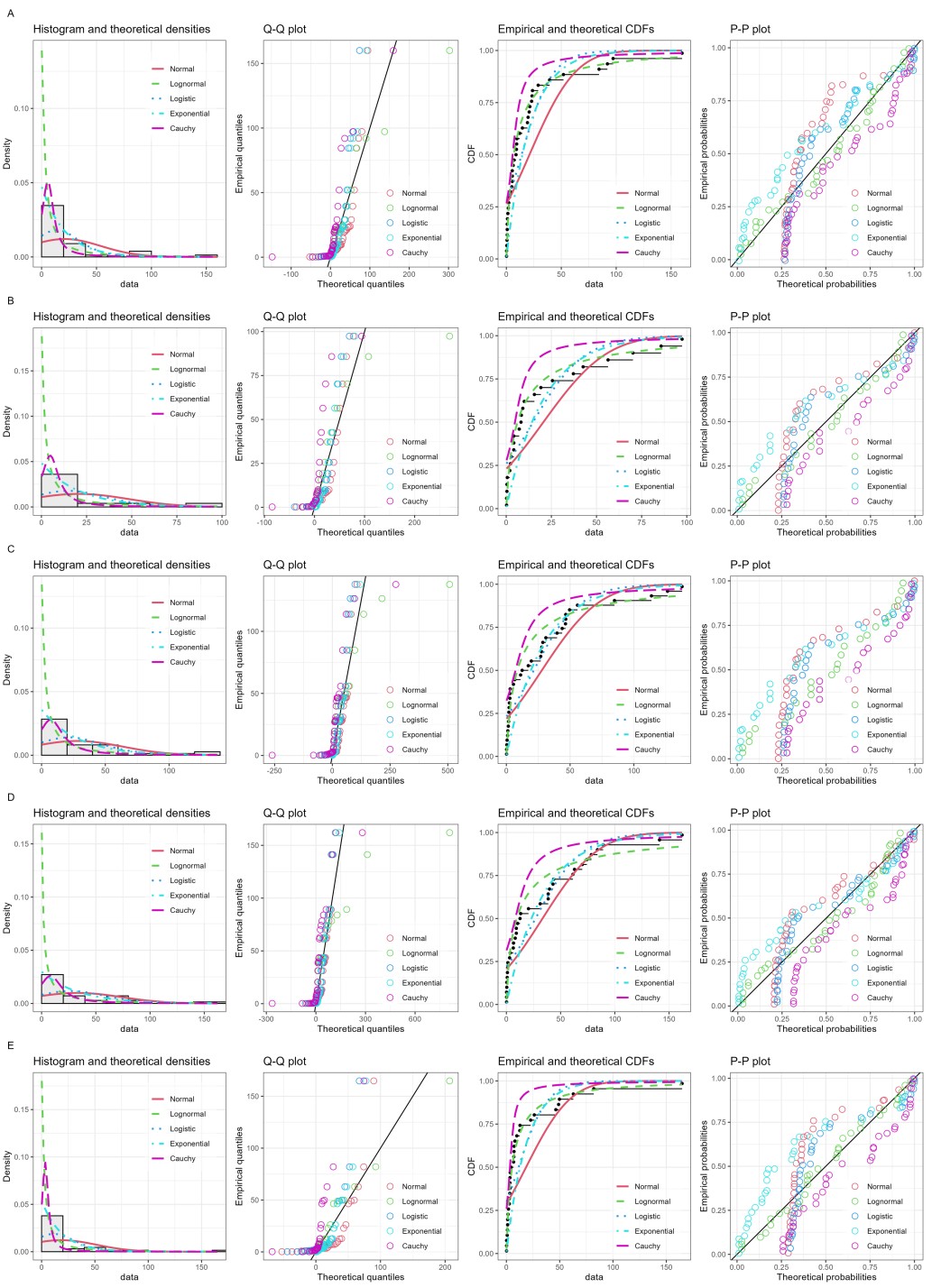

**Figure 3** Histogram, normal Q-Q, CDF and P-P plots of nonzero precipitation records in five stations of southern Thailand: (A) Songkhla (B) Songkhla-Sadao (C) Yala (D) Narathiwat (E) Pattani.

**Table 3** The AIC and BIC results for five associated models.

| Stations | Criterion | Models | | | | |
|---|---|---|---|---|---|---|
| | | Normal | Lognormal | Logistic | Exponential | Cauchy |
| Songkhla | AIC | 387.611 | 305.171 | 373.337 | 317.644 | 345.549 |
| | BIC | 390.938 | 308.498 | 376.664 | 319.308 | 348.876 |
| Songkhla-Sadao district | AIC | 241.141 | 196.707 | 238.534 | 203.226 | 225.198 |
| | BIC | 243.579 | 199.145 | 240.971 | 204.445 | 227.635 |
| Yala | AIC | 373.538 | 313.718 | 368.171 | 322.168 | 365.426 |
| | BIC | 376.760 | 316.940 | 371.393 | 323.779 | 368.648 |
| Narathiwat | AIC | 362.209 | 310.600 | 359.299 | 317.455 | 358.947 |
| | BIC | 365.320 | 313.711 | 362.410 | 319.010 | 362.058 |
| Pattani | AIC | 328.067 | 242.474 | 313.959 | 260.584 | 273.318 |
| | BIC | 331.060 | 245.467 | 316.952 | 262.080 | 276.311 |

**Table 4** Summary statistics for five stations.

| Weather stations | $i$ | $n_{i0}$ | $n_{i1}$ | $\hat{d}_i$ (%) | $\hat{\mu}_i$ | $\sigma_i^2$ | $\hat{\lambda}_i$ |
|---|---|---|---|---|---|---|---|
| Songkhla | 1 | 39 | 23 | 37.097 | 1.909 | 2.982 | 9.317 |
| Songkhla-Sadao district | 2 | 25 | 37 | 59.677 | 1.828 | 3.509 | 9.766 |
| Yala | 3 | 37 | 25 | 40.323 | 2.155 | 3.490 | 10.774 |
| Narathiwat | 4 | 35 | 27 | 43.548 | 2.253 | 4.238 | 12.411 |
| Pattani | 5 | 33 | 29 | 46.774 | 1.669 | 2.950 | 8.607 |

stations considered were overdispersed (*i.e.,* the variance was greater than the mean) and zero-inflated (Table 4). Thus, the ZILN distribution is an appropriate model for estimating parameters in the construction of SCIs for multiple comparisons between their variances.

For three populations, all of the methods produced 95% SCIs for all pairwise comparisons among variances covering the true parameter. Meanwhile, the SCI constructed *via* the Bayesian approach based on the PMB prior maintained a good balance between LER and UER and provided the narrowest AWs except for small sample sizes. On the other hand, the PB-based SCI could handle extreme cases when the sample sizes were small with large variances. For five populations, the PB-based SCI performed the best overall, with the Bayesian approach based on the RB prior for small-to-large sample sizes and the GPQ approach for medium-to-large and large sample sizes providing acceptable results, and thus can be recommended as alternative SCIs.

# APPENDIX

The proofs of the methods for constructing the SCI for $\lambda_{ik}$ are covered here.

## The GPQ approach
**Proof of Theorem 1** The proof is similar to *Hannig et al. (2006)*, *Kharrati-Kopaei & Eftekhar (2017)*, and *Maneerat, Niwitpong & Niwitpong (2021b)*. Since random variable $Q = \frac{\hat{\lambda}_{ik} - \lambda_{jk}}{\sqrt{\widehat{Var}(\hat{\lambda}_{ik})}}$ is obtained by applying the central limit theorem, where $\hat{\lambda}_{ik}$ is an estimate of

**Table 5  95% SCIs of all pairwise log-ratios of precipitation variabilities among five weather stations in lower southern Thailand.**

| Methods | Limits | All pairwise log-ratios of precipitation variabilities among weather stations | | | | |
|---|---|---|---|---|---|---|
| | | Songkhla/ Songkhla-sadao −0.4489 | Songkhla/ Yala −1.4568 | Songkhla/ Narathiwat −3.0939 | Songkhla/ Pattani 0.71043 | Songkhla-sadao/Yala −1.0079 |
| Bayesian SCIs -based PMB prior | Lower | −8.7881 | −9.796 | −11.4331 | −7.6287 | −9.3471 |
| | Upper | 7.8903 | 6.8824 | 5.2452 | 9.0496 | 7.3313 |
| | Width | 16.6783 | 16.6783 | 16.6783 | 16.6783 | 16.6783 |
| Bayesian SCIs -based RB prior | Lower | −9.4711 | −10.479 | −12.1161 | −8.3117 | −10.0301 |
| | Upper | 8.5733 | 7.5654 | 5.9283 | 9.7326 | 8.0143 |
| | Width | 18.0444 | 18.0444 | 18.0444 | 18.0444 | 18.0444 |
| SCI-based GPQ | Lower | −9.3037 | −9.2166 | −11.9695 | −6.6362 | −10.4292 |
| | Upper | 8.4059 | 6.303 | 5.7816 | 8.0571 | 8.4134 |
| | Width | 17.7096 | 15.5196 | 17.7511 | 14.6932 | 18.8426 |
| SCI-based PB | Lower | −7.4257 | −7.5709 | −10.0871 | −5.0781 | −8.4311 |
| | Upper | 6.5279 | 4.6573 | 3.8992 | 6.4989 | 6.4153 |
| | Width | 13.9536 | 12.2281 | 13.9863 | 11.577 | 14.8464 |
| Methods | Limits | Songkhla-sadao/ Narathiwat −2.645 | Songkhla-sadao/ Pattani 1.1593 | Yala/ Narathiwat −1.6371 | Yala/ Pattani 2.1672 | Narathiwat/ Pattani 3.8043 |
| Bayesian SCIs -based PMB prior | Lower | −10.9842 | −7.1798 | −9.9763 | −6.1719 | −4.5348 |
| | Upper | 5.6941 | 9.4985 | 6.702 | 10.5064 | 12.1435 |
| | Width | 16.6783 | 16.6783 | 16.6783 | 16.6783 | 16.6783 |
| Bayesian SCIs -based RB prior | Lower | −11.6672 | −7.8629 | −10.6593 | −6.855 | −5.2178 |
| | Upper | 6.3771 | 10.1815 | 7.385 | 11.1894 | 12.8266 |
| | Width | 18.0444 | 18.0444 | 18.0444 | 18.0444 | 18.0444 |
| SCI-based GPQ | Lower | −13.0047 | −7.9247 | −11.078 | −5.8532 | −5.2999 |
| | Upper | 7.7146 | 10.2433 | 7.8037 | 10.1876 | 12.9086 |
| | Width | 20.7193 | 18.168 | 18.8817 | 16.0408 | 18.2085 |
| SCI-based PB | Lower | −10.8075 | −5.9981 | −9.0757 | −4.1522 | −3.369 |
| | Upper | 5.5175 | 8.3168 | 5.8014 | 8.4866 | 10.9777 |
| | Width | 16.325 | 14.3149 | 14.8771 | 12.6388 | 14.3467 |

the log-ratio transformation of variances of the ZILNs ($\lambda_{jk}$), and $\widehat{Var}(\hat{\lambda}_{ik})$ is the approximate variance of $\hat{\lambda}_{ik}$. Consider

$$\mathrm{P}\left(\lambda_{jk} \in \left[\hat{\lambda}_{ik} \mp q_{\alpha}^{\mathrm{GPQ}}\sqrt{\widehat{Var}(\hat{\lambda}_{ik})}\right]\right) = \mathrm{P}\left(\max_{i \neq k}\left|\frac{\hat{\lambda}_{ik} - \lambda_{jk}}{\sqrt{\widehat{Var}(\hat{\lambda}_{ik})}}\right| \leq q_{\alpha}^{\mathrm{GPQ}}\right)$$
$$= \mathrm{P}\left(Q_n \leq q_{\alpha}^{\mathrm{GPQ}}\right) \qquad (33)$$

as $n \to \infty$. Hence, $\mathrm{P}\left(\hat{\lambda}_{ik} - q_{\alpha}^{\mathrm{GPQ}}\sqrt{\widehat{Var}(\hat{\lambda}_{ik})} < \lambda_{jk} < \hat{\lambda}_{ik} + q_{\alpha}^{\mathrm{GPQ}}\sqrt{\widehat{Var}(\hat{\lambda}_{ik})}\right) \to 1 - \alpha; \forall i \neq k.$

**The parametric bootstrap approach**

**Proof of Theorem 2**  The proof is similar to *Hannig et al. (2006)*, *Li, Song & Shi (2015)*; *Kharrati-Kopaei & Eftekhar (2017)* and *Maneerat, Niwitpong & Niwitpong (2021b)*. Recall

that $\widehat{Var}(\hat{\lambda}_{ik})$ is the estimated variance of $\hat{\lambda}_{ik}$. From the $100(1-\alpha)\%$ SCI for $\lambda_{ik}$ based on the PB approach, we can derive

$$P\left[\lambda_{ik} \in \left(\hat{\lambda}_{ik} \mp M_{\alpha}^{\mathrm{PB}}\sqrt{\widehat{Var}(\hat{\lambda}_{ik})}\right)\right] = P\left(\max_{i \neq k}\left|\frac{\hat{\lambda}_{ik} - \lambda_{jk}}{\sqrt{\widehat{Var}(\hat{\lambda}_{ik})}}\right| \leq m_{\alpha}^{\mathrm{PB}}\right) \tag{34}$$

Let $n_i/n \to \varphi_i$ as $n = n_1 + n_2 + \ldots + n_h \to \infty$. Thus, by applying the central limit theorem, $n(\hat{\lambda}_i - \lambda_i) \to W_i$, we arrive at

$$W = (W_1, W_2, \ldots, W_h) \sim N(0, \sigma_i^2/\varphi_i) \tag{35}$$

By applying Slutsky's theorem, we obtain

$$\max_{i \neq k}\left|\frac{\hat{\lambda}_{ik} - \lambda_{ik}}{\sqrt{\widehat{Var}(\hat{\lambda}_{ik})}}\right| \xrightarrow{d} \max_{i \neq k}\left|\frac{W_i - W_k}{\sqrt{\frac{\sigma_i^2}{\varphi_i} + \frac{\sigma_k^2}{\varphi_k}}}\right| \tag{36}$$

Assume that $W_i$ and $W_i^*$ are iid random variables, then

$$T(\mathbf{Y}, \mathbf{Y}^*, \mathbf{d}, \boldsymbol{\mu}, \boldsymbol{\sigma^2}) \to \max_{i \neq k}\left|\frac{W_i^* - W_k^*}{\sqrt{\frac{\sigma_i^2}{\varphi_i} + \frac{\sigma_k^2}{\varphi_k}}}\right|, \tag{37}$$

where $T(\mathbf{Y}, \mathbf{Y}^*, \mathbf{d}, \boldsymbol{\mu}, \boldsymbol{\sigma^2})$ comprises the distribution of a continuous random variable. Since $m_{\alpha}^{PB}(n) \to m_{1-\alpha}$, then

$$P\left(\max_{i \neq k}\left|\frac{\hat{\lambda}_{ik} - \lambda_{ik}}{\sqrt{\widehat{Var}(\hat{\lambda}_{ik})}}\right| \leq m_{\alpha}^{PB}(n)\right) \to P\left(\max_{i \neq k}\left|\frac{W_i - W_k}{\sqrt{\frac{\sigma_i^2}{\varphi_i} + \frac{\sigma_k^2}{\varphi_k}}}\right| \leq m_{1-\alpha}\right)$$

$$= P\left(\max_{i \neq k}\left|\frac{W_i^* - W_k^*}{\sqrt{\frac{\sigma_i^2}{\varphi_i} + \frac{\sigma_k^2}{\varphi_k}}}\right| \leq m_{1-\alpha}\right)$$

$$= 1 - \alpha \tag{38}$$

as $a \to \infty$, where $m_{1-\alpha}$ denotes the $100(1-\alpha)^{th}$ percentile of $\max_{i \neq k}\left|\frac{W_i^* - W_k^*}{\sqrt{\frac{\sigma_i^2}{\varphi_i} + \frac{\sigma_k^2}{\varphi_k}}}\right|$. This implied that

$$P\left[\lambda_{ik} \in \left(\hat{\lambda}_{ik} \mp M_{\alpha}^{\mathrm{PB}}\sqrt{\widehat{Var}(\hat{\lambda}_{ik})}\right)\right] \to 1 - \alpha.$$

### Funding
This research was funded by King Mongkut's University of Technology North Bangkok. (Grant No. KMUTNB-65-KNOW-09). The funders had no role in study design, data collection and analysis, decision to publish, or preparation of the manuscript.

## Grant Disclosures

The following grant information was disclosed by the authors:
King Mongkut's University of Technology North Bangkok: KMUTNB-65-KNOW-09.

## Competing Interests

The authors declare there are no competing interests.

## Author Contributions

- Patcharee Maneerat conceived and designed the experiments, performed the experiments, analyzed the data, prepared figures and/or tables, authored or reviewed drafts of the paper, and approved the final draft.
- Sa-Aat Niwitpong conceived and designed the experiments, analyzed the data, authored or reviewed drafts of the paper, and approved the final draft.

## Data Availability

The raw data and R code are available in the Supplementary File.

## Supplemental Information

Supplemental information for this article can be found online at http://dx.doi.org/10.7717/peerj.12659#supplemental-information.

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
