# Peer review of "Multiple comparisons of precipitation variations in different areas using simultaneous confidence intervals for all possible ratios of variances of several zero-inflated lognormal models"

_PeerJ, doi:10.7717/peerj.12659_

## Round 0.1 · original submission · Minor Revisions

This manuscript has been reviewed by two experts in this research field and both of them suggest a minor revision. Please modify your manuscript based on these comments accordingly.

Reviewer 1 ·

Basic reporting

The authors clearly state the objectives, methodology and present results in a lucid way.
Overall the quality of the manuscript conform to the standards of PeerJ. However, the novelty of the study is questionable as the authors themselves have published similar kind of studies previously. Hence, the authors need to highlight the research gaps and need for this study. Multiple comparisons across different study areas wouldn't serve the novelty purpose.

Experimental design

Justify with reasons for selecting Bayesian, generalized pivotal quantity and parametric bootstrap approaches for formulating SCI.
Line 79: what "Model" is that?
Table 2: I think the data period seems missing in this table. Mention the dates or week in the table.

Validity of the findings

Discuss the limitations of the GPQ method and the reasons for it.
The results could be further analyzed and reported in the form of text.

Additional comments

Language needs a thorough check. Several typo errors were seen during my review. Example line 106: generalized instead of generalize

Reviewer 2 ·

Basic reporting

The paper is written well and clear.

Experimental design

The methods were designed correctly and had sufficient information.

Validity of the findings

The new knowledge was proposed in this paper.

Additional comments

There are some comments for improving the paper:
1. The results must be added in the abstract.
2. What is the difference between the zero-inflated lognormal (ZILN) model and the delta-lognormal model? Please add it in the paper.
3. Many formula sentences are not complete (lack of the full stop). See Eqs.(1), (3), (4), (5), (8), (9), (20), (22).
4. In lines 85-89, the symbols and are used inconsistency.
5. In Table 5, why do the widths of Bayesian SCIs -based PMB prior and Bayesian SCIs -based RB prior are the same in all pairwise?

---

## Round 0.2 · accepted · Accept

Thanks for your revision based on review comments, and I am happy to accept it in current form. Congratulation!

I hope you will submit your manuscript again to our Journal in the future.